# A Novel Method to Prepare Transparent, Flexible and Thermally Conductive Polyethylene/Boron Nitride Films

**DOI:** 10.3390/nano12010111

**Published:** 2021-12-30

**Authors:** Mingming Yi, Meng Han, Junlin Chen, Zhifeng Hao, Yuanzhou Chen, Yimin Yao, Rong Sun

**Affiliations:** 1School of Chemical Engineering and Light Industry, Guangdong University of Technology, Guangzhou 510006, China; 203701063@csu.edu.cn (M.Y.); 2112106060@mail2.gdut.edu.cn (J.C.); m15521162019@163.com (Y.C.); 2Shenzhen Institute of Advanced Electronic Materials, Shenzhen Institute of Advanced Technology, Chinese Academy of Sciences, Shenzhen 518055, China; meng.han@siat.ac.cn (M.H.); ym.yao@siat.ac.cn (Y.Y.); rong.sun@siat.ac.cn (R.S.); 3Shenzhen College of Advanced Technology, University of Chinese Academy of Sciences, Shenzhen 518055, China

**Keywords:** polyethylene, BN nanoplates, electrostatic self-assembly, thermal conductivity, translucent

## Abstract

The high thermal conductivity and good insulating properties of boron nitride (BN) make it a promising filler for high-performance polymer-based thermal management materials. An easy way to prepare BN-polymer composites is to directly mix BN particles with polymer matrix. However, a high concentration of fillers usually leads to a huge reduction of mechanical strength and optical transmission. Here, we propose a novel method to prepare polyethylene/boron nitride nanoplates (PE/BNNPs) composites through the combination of electrostatic self-assembly and hot pressing. Through this method, the thermal conductivity of the PE/BNNPs composites reach 0.47 W/mK, which gets a 14.6% improvement compared to pure polyethylene film. Thanks to the tight bonding of polyethylene with BNNPs, the tensile strength of the composite film reaches 1.82 MPa, an increase of 173.58% compared to that of pure polyethylene film (0.66 MPa). The fracture stress was also highly enhanced, with an increase of 148.44% compared to pure polyethylene film. Moreover, the addition of BNNPs in PE does not highly reduce its good transmittance, which is preferred for thermal management in devices like light-emitting diodes. This work gives an insight into the preparation strategy of transparent and flexible thermal management materials with high thermal conductivity.

## 1. Introduction

Thermal management is essential to the performance and lifetime of modern electronic devices, such as chips, solar cells, and light-emitting diodes (LEDs), especially considering their fast miniaturization and integration. In the past decades, polymers have been explored as excellent materials for electronic devices packaging due to their low density, ease of processing, and excellent dielectric properties [1,2]. With an extremely high transparency, polyethylene (PE) shows good chemical stability and insulation performance, which is very suitable for the thermal management of light-emitting diodes [3,4,5]. Unfortunately, pure PE film cannot meet the urgent heat dissipation requirement due to its low intrinsic thermal conductivity (below 0.4 W/mK) [6,7]. A general way to improve its thermal conductivity is to introduce thermally conductive fillers, such as Al_2_O_3_, ZnO, and boron nitride (BN), into PE film [7,8]. However, due to the scattering of light at the filler-matrix interfaces, the introduction of these fillers often highly reduces the transmittance of PE and narrows its application in devices like high-output light-emitting diodes [9,10]. Therefore, novel technologies that can highly enhance the heat dissipation ability of transparent polymers like PE, yet still maintain its high flexibility and transmittance, are still in urgency.

As an excellent 2D material with ultrahigh thermal conductivity and good insulating properties, BN has attracted tremendous attention in the development of highly thermal conductive composites [6,7,11,12]. However, there are scarcely functional groups at the surface of BN, which makes it difficult to be wedded by polymer matrix and thus prevents its further dispersion [5]. Surface functionalization has proved to be an effective way to deal with such a problem, however, it will inevitably lead to the sacrifice of its intrinsic thermal conductivity [7,13,14,15]. Constructing a 3D BN network by ice-template paves another way to overcome these wedding and dispersion problems. Through infiltration of the polymer into the network, composites with high through-plane thermal conductivity were obtained [16,17]. However, this method is complicated and the network is very brittle, which limits its large-scale application [16]. Until now, finding simple ways to prepare polymer/BN composites with high thermal conductivity was challenging.

In recent years, thanks to the high surface energy of nanomaterials brought about by the size effect, electrostatic self-assembly was introduced to prepare composites with different functional properties [18,19,20]. For instance, Fu et al. used strong electrostatic force to facilitate the surface of WO_3_ nanosheet to mix with bovine serum albumin [21]. Zhang et al. fabricated a sandwich-like composite with CoAl-LDH nanoplates electrostatically assembled on both sides of polypyrrole/graphene substrate [22]. Nie et al. introduced electrostatic self-assembly into the preparation of g-C_3_N_4_/ZnO composites and realized a uniform distribution of g-C_3_N_4_ nanosheets on the ZnO microspheres surfaces [23]. However, it is worthwhile to mention that the above process of electrostatic self-assembly is very harsh and sometimes dangerous [24].

Here in the current work, we reported on a facile electrostatic self-assembly method to prepare PE/BNNPs composite film with improved thermal conductivity and dielectric properties, but without highly sacrificing the flexibility and transmittance of PE. Thanks to the high volume resistivity of polyethylene(>10^14^ Ω·m), electrostatic charge at the film surface will not leak out for a long time (more than three months) [25]. PE film with corona treatment possesses a large number of electrostatic charges and thus avoids the harsh experimental condition of high charge level like those in the process of traditional electrostatic self-assembly. Under the action of electrostatic force, BNNP was evenly distributed on the surface of the PE film and could further prepare the composite material. This method provides a new strategy for the preparation of polymer composites with high thermal conductivity and optical transparence.

## 2. Experimental Section

### 2.1. Materials

The BNNPs with an average size of 2 µm were purchased from Denka Co., Ltd., Tokyo, Japan. The PE film with a thickness of 9 µm was purchased from Clorox Co., Ltd., Oakland, CA, USA. The deionized water was obtained from a Milli-Q system (Millipore, Billerica, MA, USA). All the reagents were of analytical grade and used as received.

### 2.2. Methods

#### 2.2.1. Preparation of BNNPs Film in Air-Water Interface

100 mg BNNPs were slowly added to 500 mL deionized water. After that, the solution was stirred at 550 rpm for 24 h, allowing large boron nitride particles to settle down. Then, the solution was continually stirred at 200 rpm for 2 h in order for uniform distribution of the BNNPs at the surface of the DI water.

#### 2.2.2. Preparation of Single-Layered PE/BNNPs Film

The single-layered PE/BNNPs film was produced by electrostatic attraction of the floating BNNPs film on the PE film. A polytetrafluoroethylene mold was first used to support the flexible PE film, then the supported film was slowly moved close to the surface of the floating BNNPs film, and finally get in touch with it. After that, the mold and film were slowly lifted up from the container, and a uniform BNNPs film formed at the surface of the PE film. The film was then dried at 70 °C for 2 h and peeled off from the polytetrafluoroethylene mold.

#### 2.2.3. Preparation of Multilayered PE/BNNPs Film

The obtained single-layered PE/BNNPs film was further assembled into a multilayered film through layered stacking and hot-pressing. Five single-layered PE/BNNPs films with the same lateral sizes were stacked layer by layer and then hot-pressed at 10 MPa and 120 °C for 2 h, during which the PE film melted and attached together. Then, the film cooled down slowly while the pressure was kept the same, and the multilayered PE/BNNPs films formed.

### 2.3. Characterization

The morphological analysis of the film was conducted by FEI Nova Nano-SEM 450 field emission scanning electron microscopy (FE-SEM) (FEI, Hillsboro, OR, USA) with 5 kV acceleration voltage. The tensile strain in mechanical property was analyzed by a universal stretching machine (Shimadzu AG-X Plus100N, Kyoto, Japan). Atomic force microscopy (AFM) images were taken in the tapping mode with a Bruker Dimension Icon apparatus (Rheinstetten, Germany). The variations of the surface temperature of PE/BNNPs composites were recorded by an infrared thermograph (FLIR T1040) (FEI, Hillsboro, OR, USA).

The in-plane thermal diffusivity was first measured by the transient electrothermal (TET) technique. Then, the thermal conductivity was calculated by using the following equation:(1)$$k=\alpha \rho c_{p}$$
where *ρ* is the density of the composites, *c_p_* is the specific heat obtained from the differential scanning calorimetry (DSC, TA Q2000) (TA, Newcastle, DE, USA), and *α* is the in-plane thermal diffusivity. In the TET measurement, a small piece of film with a width of about 100 µm and length of 1 mm was prepared. A thin gold film with a thickness of about 10 nm was sputtering coated on the sample to make it electrically conductive. The gold coating performed as an electric heater as well as thermal sensor, to heat the sample and monitor the temperature variation during the TET measurement. Typically, a square wave electric current is fed to the gold coating, and its electric resistance and voltage change is recorded, which mimics the temperature changes of the sample. By fitting the voltage change curve with a theoretical thermal transport model, the thermal diffusivity of the sample is extracted. Details of the TET technique can be found in reference [26,27].

Thermo-gravimetric analyses (TGA) of the samples were performed in a TGA2 thermo-analyzer (Mettler Toledo, Zurich, Switzerland) in the temperature range of 80–800 °C with a heating rate of 10 °C/min and an air flow of 50 mL/min. The breakdown strength was analyzed by the CS9912BX AC/DC Hipot Tester (ChangSheng, Nanjing, China). Calculation of the specific breakdown voltage follows the formula below:(2)$$E=e^{\left|\frac{b}{\beta }\right|}$$
where *E* is the breakdown voltage of different samples, *e* is the natural logarithm, *b* and *β* are the intercept and slope of linear function of ln[−ln(1 − p)] versus ln*E*, respectively.

## 3. Results and Discussion

Our designed fabrication procedure of PE/BNNPs film involves the molding of BNNPs film, the electrostatic self-assembly of PE/BNNPs film, and hot pressing, as depicted in Figure 1. In a typical process, the mixture of BNNPs and deionized water was stirred for 15–30 min to form a dense and homogeneous Langmuir film at the air-water interface. The single-layered PE/BNNPs film was obtained through electrostatic self-assembly of BNNPs on PE film. With a large number of negative charges at the surface of PE film, the BNNPs film at the air-water interface was attracted and bonded to the PE film rapidly. Thus, a single-layered PE/BNNPs film composed of a layer of PE film and a layer of uniformly-distributed BNNPs was obtained. Such a composite film is easy to be damaged, which motivated us to seek different ways to enhance its stability and mechanical properties. The multilayered PE/BNNPs films were then fabricated via hot pressing of several such single-layered PE/BNNPs films. The BNNPs loadings in the single-layered PE/BNNPs film and multilayered PE/BNNPs film were 1.6 wt% and 2.3 wt%, respectively, as confirmed by TGA (Appendix A).

The surface morphology of the prepared single-layered PE/BNNPs film was characterized by SEM. Figure 2a,b shows the microstructure of the single-layered PE/BNNPs film, from which one can see that the BNNPs lie on the surface of PE film in stacking form. Thanks to the powerful electrostatic force, the BNNPs were tightly bonded to the surface of PE film and exhibit a distinct layered structure. The edge of PE film was not completely covered by BNNPs, mainly due to the highly reduced electrostatic force at the edge of film. Furthermore, electrostatic charges at the edge of the PE film may fade out more easily than those at the center portion. The single-layered PE/BNNPs film was further characterized by AFM, as shown in Figure 2c,d. The BNNPs were stacked at the PE surface with relatively good uniformity. More details about the BNNPs stackings are shown in Appendix A.

To serve as thermal management materials for electronics, such as high-out LEDs, the prepared film should exhibit relatively high thermal conductivity. However, owing to the very thin thickness and high flexibility of the prepared films, measurement of its in-plane thermal conductivity through the commonly used laser flash method is challenging. Instead, a transient electrothermal technique (TET) was used to first extract the in-plane thermal diffusivity of the prepared films. Then, the thermal conductivity can be easily calculated as: $k=\alpha \cdot \rho c_{p}$. Figure 3a shows the raw data and the corresponding fitting curve in the TET measurement of the multilayered PE/BNNPs film sample. Through the calculation of the original data by matlab software, the thermal diffusivity of the sample was obtained (Appendix A). Then, with the further characterization of the film density and the specific heat, the in-plane thermal conductivity was calculated (Figure 3b). With the addition of BNNPs, the in-plane thermal conductivity gained a moderate improvement due to the formation of heat conduction pathways caused by the uniformed distribution of BNNPs during the electrostatic assembly process. After hot pressing, the tight connection between BNNPs and PE film further enhanced the thermal conductivity. Due to the effect of Umklapp phonon-phonon scatterings, the thermal conductivity of all the samples decreased with the increased temperature, which agrees well with the previous report [28].

The optical transmission performance of the prepared PE/BNNPs films was then studied. As shown in Figure 4, the addition of BNNPs does not cause a large drop in the optical transmittance. It is mostly due to the uniform distribution of BNNPs on the surface of PE film during the electrostatic assembly process. Different films supported on a paper frame are shown in Figure 4a–c, where we can see the single-layered and multilayered PE/BNNPs films are still of good transmittance. We test the optical transparency of different films and compare the transmittance of each film at around 550 nm, where the human eye is most sensitive. As shown in Figure 4d, the pure PE film has an ultrahigh light transmittance (larger than 90% in most of the wavelength range). For comparison, the transmittances of single-layered and multilayered PE/BNNPs films were lower, but still high enough to be transparent. Appendix A provided more information of the transparency and flexibility of the single-layered PE/BNNPs film. After the layered stacking and hot-pressing, the transmittance reduced a little further (about 30%), which is mainly due to the partial aggregation of BNNPs.

The thermal and optical properties, and also the mechanical properties, of the prepared PE/BNNPs films greatly influence their practical applications. Figure 5a shows the strain–stress curve of different films, where the strain and stress clearly increase from pure PE film to single-layered PE/BNNPs film and to Multilayered PE/BNNPs film. Figure 5b shows that the shape variable of multilayered PE/BNNPs film can reach a staggering 683%. From Figure 5c,d, the tensile modulus of the single-layered PE/BNNPs film reaches 1.06 MPa and have increased 59.61%, while the multilayered PE/BNNPs film reaches 1.82 MPa and have increased 173.58%, compared to pure PE film. It is worth mentioning that due to the tight bonding of the PE layer and BNNPs, the fracture stress of multilayered PE/BNNPs films can reach 20.39 MPa, with a 148.44% improvement compared to pure PE film. In the condition of low tensile strength, the tightly combined BNNPs with PE layer can prevent the deformation of the film. With the increasing tensile strength, the soft PE layers guarantee no brittle fracture but only deformation happens. When the tensile strength continues increasing to the deformation limit of PE layer, the BNNPs layers guarantee extremely tough properties to prevent fracture.

Moreover, the insulating property of the film has been largely improved with the addition of BNNPs in comparison with pure PE film. By linear fitting of ln(−ln(1 − p))) versus ln E, b, and *β* of the prepared single-layered BNNPs/PE film and multilayered BNNPs/PE film, the pure PE was extracted (Appendix A). Then, the electrical breakdown strength of the three different samples were calculated with Equation (2). As shown in Figure 6, the breakdown strength of multilayered PE/BNNPs films reaches 106.30 KV/mm, with an 87.0% increase compared to 56.85 KV/mm of pure PE film. BNNPs on and in the PE film block the transmission of electrons in the film due to the ultrahigh volume resistivity of pure h-BN (>10^13^ Ω·cm). The improved insulating property increases application prospects in electronic devices.

The heat dissipation performance of the prepared films were further characterized by using an infrared camera. As shown in Figure 7a, a copper stick is placed on the hot plate and serves as a small heat source. A specific structure was built to suspend the films and attached the center of the film to the end of the copper stick. The temperature of the films were monitored by an infrared camera. Figure 7b shows the temperature evolution at the edge of the suspend films, where multilayered PE/BNNPs film shows much faster temperature rise compared to the BNNPs/PE film PE/BNNPs film and the pure PE film, meaning it has higher thermal diffusivity. Moreover, the steady-state temperature distribution and the highest temperature of the suspended films were extracted (as shown in Figure 7c–e). Clearly, the highest temperature of multilayered PE/BNNPs was over 1.5 °C lower than those of single-layered PE/BNNPs film and pure PE film, which confirmed its higher thermal conductivity.

## 4. Conclusions

A new thermally conductive and transparent material with excellent mechanical properties was prepared through electrostatic self-assembly and hot pressing. The thermal conductivity of the prepared multilayered PE/BNNPs film with a 2.3 wt% loading of BNNPs was 0.47 W/mK at 310 K, an increase of 14.6% compared to that of pure PE film. Meanwhile, the tensile modulus and the fracture stress were also enhanced by 173.58% and 148.44%, respectively. The electric breakdown strength was highly enhanced, due to the large band gap of BN and its uniform distribution in the prepared composite. The method of electronic self-assembly in normal temperature and pressure is a very promising approach for the preparation of composites without complicated functionalization. Further attention will be paid to the accurate control of the content of BNNPs. Our research provides a novel way for the preparation of polymer composites with high thermal conductivity, flexibility and transmittance for the thermal management of specific electronic packaging.

## Figures and Tables

**Figure 1 nanomaterials-12-00111-f001:**
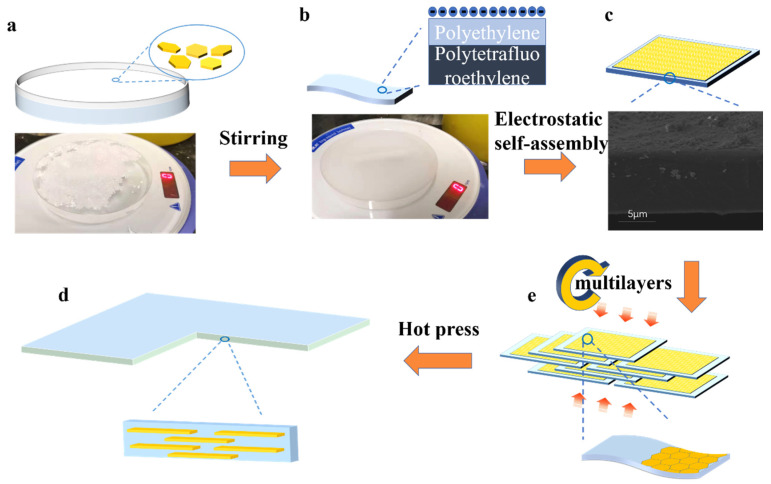
Schematic of the preparation process of the PE/BNNPs films. (**a**) BNNPs added into deionized water. (**b**) BNNPs formed a thin film at the water/air interface after magnetic stirring. (**c**) Single-layered PE/BNNPs film and the corresponding cross-plane SEM image. (**d**) Stacking of single-layered PE/BNNPs film with excellent flexibility. (**e**) The multilayered PE/BNNPs film through hot-pressing.

**Figure 2 nanomaterials-12-00111-f002:**
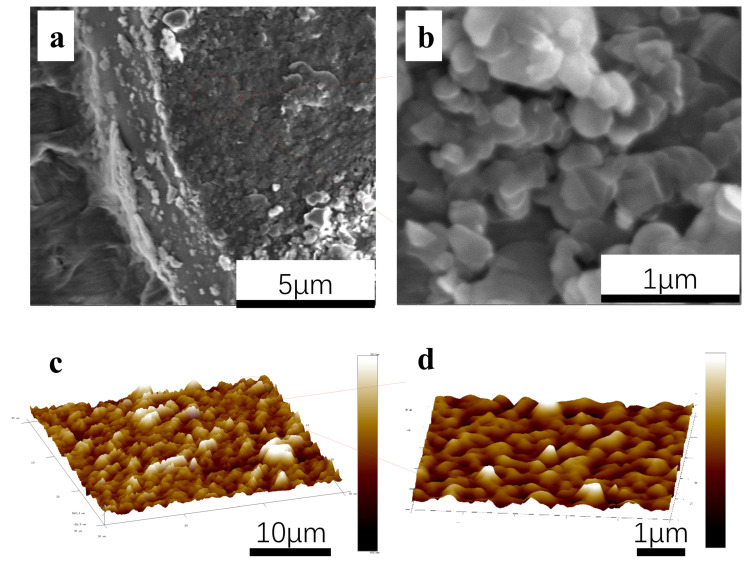
Microstructure of the single-layered PE/BNNPs film. (**a**,**b**) The in-plane FE-SEM image of stacked BNNPs on the surface of PE film; (**c**,**d**) AFM images at the centre of the single-layered PE/BNNPs film.

**Figure 3 nanomaterials-12-00111-f003:**
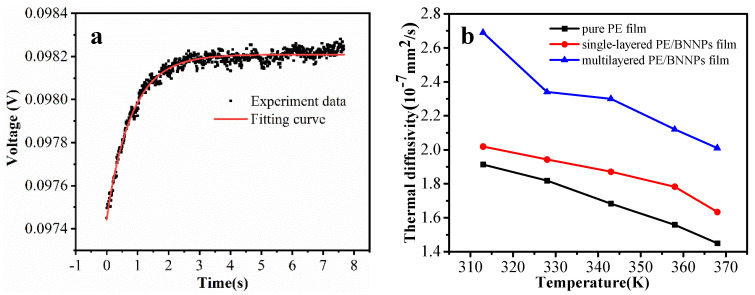
Thermal conductive property of different films. (**a**) Experiment data and fitting curve in the TET measurement of multilayered PE/BNNPs film sample. (**b**) Calculated in-plane thermal conductivity of different samples.

**Figure 4 nanomaterials-12-00111-f004:**
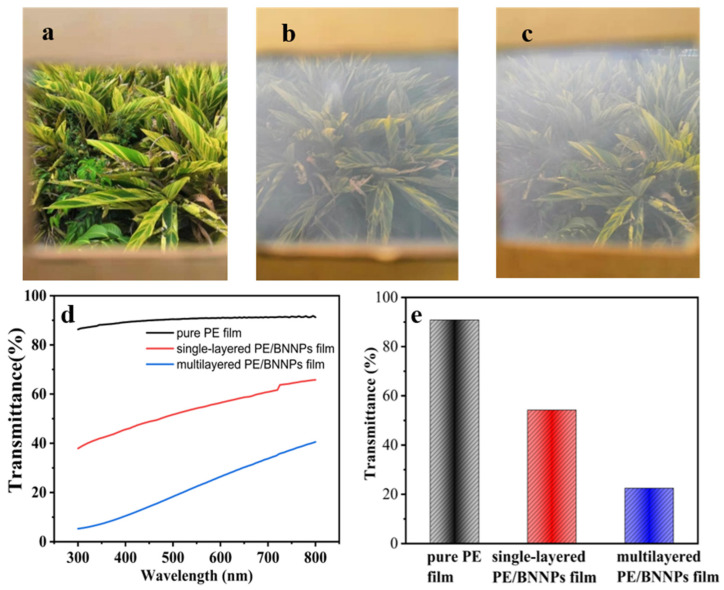
The light transmission performance of single-layered PE/BNNPs films and multilayered PE/BNNPs film. (**a**–**c**) Optical picture of pure PE film, single-layered PE/BNNPs film, and multilayered PE/BNNPs film, respectively. (**d**) The transmittance-wavelength curves of different films. (**e**) The transmittance comparison of different films at the wavelength of 550 nm.

**Figure 5 nanomaterials-12-00111-f005:**
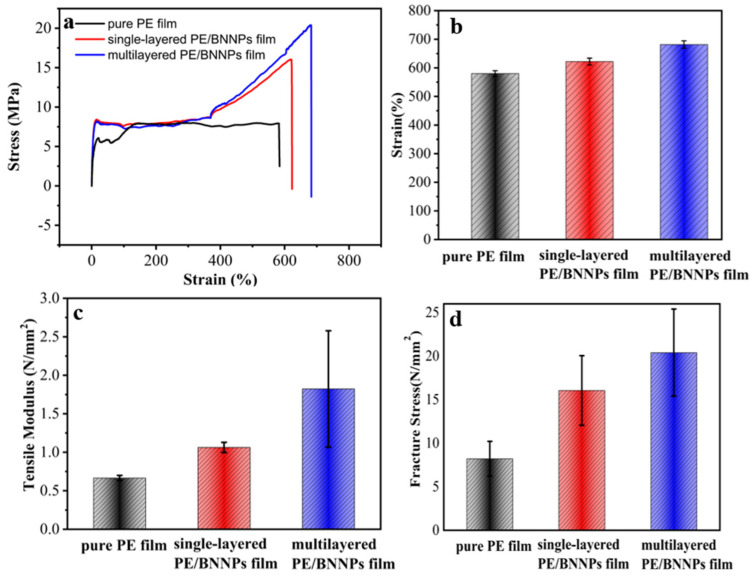
The mechanical property of pure PE film, single-layered PE/BNNPs film, and multilayered PE/BNNPs film. (**a**) The stress–strain curve of different films. (**b**) The maximum strain comparison of different films. (**c**) The tensile modulus comparison of different films. (**d**) The comparison of different films in fracture stress.

**Figure 6 nanomaterials-12-00111-f006:**
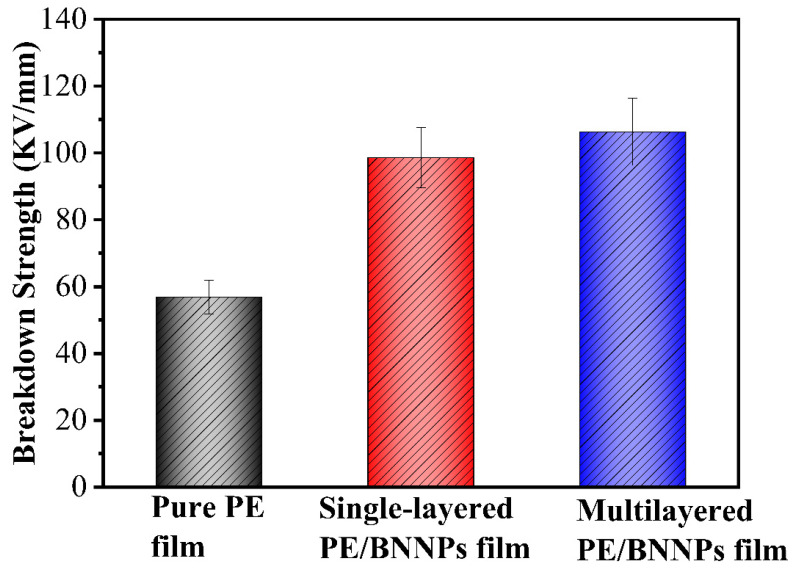
Breakdown strength of pure PE film, single-layered PE/BNNPs film, and multilayered PE/BNNPs film.

**Figure 7 nanomaterials-12-00111-f007:**
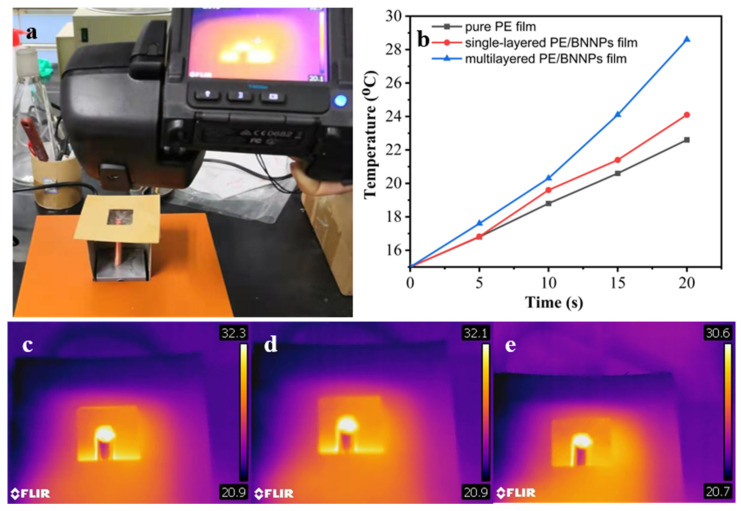
Infrared thermography characterization of the samples to confirm the improvement of in-plane thermal conductivity with the addition of BNNPs intuitively. (**a**) The experimental setup for the infrared thermography. (**b**) Temperature rise at the edge of different films. (**c**–**e**) Steady-state temperature distribution of pure PE film, single-layered PE/BNNPs film, and multilayered PE/BNNPs film, respectively.

## Data Availability

Not applicable.

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
