# Peer review of "A Novel Method to Prepare Transparent, Flexible and Thermally Conductive Polyethylene/Boron Nitride Films"

_nanomaterials, 2021, doi:10.3390/nano12010111_

Round 1

Reviewer 1 Report

The authors of this paper have presented a novel method to prepare transparent, flexible, and thermally conductive Polyethylene/Boron Nitride films. Most parts of this paper (design of the research, experimental methods, results, and discussion) are well explained. I have a few minor comments to be considered by authors for improvements.

1) In abstract section, the boron nitride nanoplates are abbreviated to BNNS in line 18, 21, 24. However, BNNP is used throughout the main article instead of BNNS. I think it is better to use BNNP in the abstract.

2) In line 136, Fig. 4a is not right reference. I think it should be Fig. 3a.

3) In Figure 1, the photo images in a, b and the SEM image in c are too small and not clear. I recommend to increase the size of images and make them clear.

4) This paper is focusing on the novel preparation method of the film, so I think it is better to describe the preparing steps in a more detailed manner. For example, you are mentioning PE film with corona treatment, but it is not described at all in the manuscript. You are using PTFE mold, but it is difficult to understand how the mold is working with the PE film for the whole process. 

5) In line 213, Fig. S5 is not a right reference. 

6) I think Fig. 3b and Fig. S3 are basically same experimental data, but the temperature points are a little different. For example, the highest temperature is about 373K in Fig. 3b, but it is about 368K in Fig. S3. Can you double check the graphs?

7) There are some typos and grammatical errors. For example, stead->steady line 269, partially->partial line 215, detailly->in detail line 108. I highly recommend authors double check these errors throughout the manuscript.

8) In line 261-265 and Fig. 7c,d,e, you are using the highest temperature points to ensure the higher thermal conductivity of the composite film. I think this is not clear because the copper rod should have same temperature for each case. Can you please explain this more clearly?    

Author Response

Response to Reviewer 1 Comments

Point 1: In abstract section, the boron nitride nanoplates are abbreviated to BNNS in line 18, 21, 24. However, BNNP is used throughout the main article instead of BNNS. I think it is better to use BNNP in the abstract.

Response 1: Thanks for your kind remind, the expression BNNS in the abstract has been replaced with BNNP.

Point 2: In line 136, Fig. 4a is not right reference. I think it should be Fig. 3a.

Response 2: Thanks for your kind remind, it is a clerical error and has been revised.

Point 3: In Figure 1, the photo images in a, b and the SEM image in c are too small and not clear. I recommend to increase the size of images and make them clear.

Response 3: Thanks for your kind remind, Figure 1 has been replaced with a higher resolution.

Point 4: This paper is focusing on the novel preparation method of the film, so I think it is better to describe the preparing steps in a more detailed manner. For example, you are mentioning PE film with corona treatment, but it is not described at all in the manuscript. You are using PTFE mold, but it is difficult to understand how the mold is working with the PE film for the whole process.

Response 4: Thanks for your kind remind, since we need to perform surface electrostatic treatment on the PE film, we purchased a corona-treated PE film for our experiment. The specific process of corona is not within the scope of our experiment, so it is not written into the experimental process. PTFE mold itself does not react with the PE film, but it can ensure that the PE film will not be deformed due to electrostatic force during the entire electrostatic self-assembly process.

Point 5: In line 213, Fig. S5 is not a right reference.

Response 5: Thanks for your kind remind, it is a clerical error and has been modified to Fig. S4.

Point 6: I think Fig. 3b and Fig. S3 are basically same experimental data, but the temperature points are a little different. For example, the highest temperature is about 373K in Fig. 3b, but it is about 368K in Fig. S3. Can you double check the graphs?

Response 6:

Thanks for your kind remind. We have double checked the raw data. The highest temperature should be the same of 368 K, as the thermal conductivity is calculated from the thermal diffusivity data. We have replaced Fig. 3b in the main manuscript, accordingly.

Point 7: There are some typos and grammatical errors. For example, stead->steady line 269, partially->partial line 215, detailly->in detail line 108. I highly recommend authors double check these errors throughout the manuscript.

Response 7: Thanks for your kind remind, we double checked these errors and revised them.

Point 8: In line 261-265 and Fig. 7c, d, e, you are using the highest temperature points to ensure the higher thermal conductivity of the composite film. I think this is not clear because the copper rod should have same temperature for each case. Can you please explain this more clearly?

Response 8: Our statistical infrared thermal imaging data in Fig. 7b is not the temperature at the center of the film but at the edge of the film. In Fig. 7c, d, e, the greater the color difference between the center and the edge of the film, the slower the heat dissipation speed. The temperature of the copper pillar and the temperature of the center of the film are indeed the same (Different from the highest scale temperature, the temperature in the center of the copper pillar is around 30 degree Celsius as shown in Fig. 7c, d, e). It is precisely because the temperature of the center of the film is the same that we can guarantee a single variable to observe the temperature of different film edges to judge the difference in the in-plane thermal conductivity of different films.

Reviewer 2 Report

Generally, presentation of this work looks good. 

  1. The measured inplane thermal conductivity is quite small.  Only slight enhancement of inplane TC was detected.
  2. This work shows improved mechanical properties with small decrease in transparency. Suggestion of suitable application of this material is required. As a thermal management material, the product of this work is much below the required level of materials properties.

Author Response

Response to Reviewer 2 Comments

Point 1: The measured inplane thermal conductivity is quite small. Only slight enhancement of inplane TC was detected.

Response 1: Thanks for your kind review. Even though the composite we prepared is improved compared to the pure PE film, it is still not high. This is due to the current technological limitation that the content of boron nitride filler is very small (2.3 wt%). In the future, continuous improvement of its process can increase the content of boron nitride filler, thereby promoting the formation of more thermally conductive paths and improving thermal conductivity of composite.

Point 2: This work shows improved mechanical properties with small decrease in transparency. Suggestion of suitable application of this material is required. As a thermal management material, the product of this work is much below the required level of materials properties.

Response 2: Thanks for your kind review. Transparency and mechanical properties are necessary properties for LED chips. Although the composite prepared by other researchers can obtain higher thermal conductivity, it is difficult to maintain a certain degree of transparency and higher mechanical properties after adding thermally conductive fillers. Therefore, the composite material we studied is expected to become a candidate for a new generation of thermal management materials for LED chips.

Reviewer 3 Report

This paper describes a new method for the preparation of polymer composites based on Polyethylene/Boron Nitride Films.  Albeit the idea and the general approach are compelling, there are a few points that should be addressed by the authors before the manuscript could be accepted for publication.

  1. Abstract

The letter font size is not uniform.

  1. Introduction

-Review the sentence in line 36 “and high electrical dielectric property.”

- The sentences from line 77 to 83 “Through this method….devices packaging.” should be moved to the conclusions section.

  1. Results and Discussion

-What is the thickness of the single and multilayered PE/BNNPs film?

- The caption of figure S3 says "the fitted in-plane thermal diffusivity of different samples.", but no fitting to the experimental results was made.

- Line 191: Figure S3

- Review the statement in line 204: “the addition of BNNPs does not cause large drop in the optical transmittance”. It was observed that the transmittance of the pure PE film at 550 nm is about 90% and in the PE/BNNPs films drops to about 25%, which represents a large drop of transmittance.

- Line 245: from the SI document these figures are labelled as S5-S7. Also, the fitting of this figures is not well performed. The data of figure S5 can not be a linear fitting.

- What can you say about the thermal transport mechanism within the composite? Is it possible to be through BNNP percolation network?

  1. Conclusions

This section is limited in both novel insights and comparative performance with related approaches. Update the references of the paper.

Author Response

Response to Reviewer 3 Comments

Point 1: The letter font size is not uniform.

Response 1: The letter font size has been revised.

Point 2: Review the sentence in line 36 “and high electrical dielectric property.”

Response 2: The description of dielectric properties in the abstract is replaced by “and excellent dielectric property.”

Point 3: The sentences from line 77 to 83 “Through this method….devices packaging.” should be moved to the conclusions section.

Response 3: The sentences from line 77 to 83 “Through this method….devices packaging.” appeared in the introduction is indeed slightly improper, it has now been replaced with more appropriate words.

Point 4: What is the thickness of the single and multilayered PE/BNNPs film?

Response 4: By measuring with a spiral micrometer, the thickness of the single and multilayered PE/BNNPs film are determined as 12 μm and 37 μm, respectively.

Point 5: The caption of figure S3 says "the fitted in-plane thermal diffusivity of different samples.", but no fitting to the experimental results was made.

Response 5: The expression of “fitted” is to express that the thermal diffusivity data in Fig. S3 are extracted from fitting process. To avoid such ambiguity, we have changed the expression as “the calculated in-plane thermal diffusivity”.

Point 6: Line 191: Figure S3

Response 6: This is a clerical error, it has been revised.

Point 7: Review the statement in line 204: “the addition of BNNPs does not cause large drop in the optical transmittance”. It was observed that the transmittance of the pure PE film at 550 nm is about 90% and in the PE/BNNPs films drops to about 25%, which represents a large drop of transmittance.

Response 7: Due to the opacity of boron nitride itself, it is difficult to avoid the drop in transparency with its addition to the PE film. After electrostatic self-assembly, the uniform distribution of boron nitride enables it to maintain a transparency of more than 50%, which is a level that is difficult to achieve with other assembly methods. After hot-pressing, the transparency is further reduced due to the change of the internal lattice structure of the PE and the reconstruction. As shown in Fig. 4c, it still has a certain degree of transparency.

Point 8: Line 245: from the SI document these figures are labelled as S5-S7. Also, the fitting of this figures is not well performed. The data of figure S5 can not be a linear fitting.

Response 8:

Thanks for your kind suggestion. We agree that the data in figure S5 seems not that linear. However, from the principle of the CS9912BX AC/DC Hipot Tester, ln(-ln(1-p)) versus lnE should be linear. We attribute the deviation of the raw data from linear distribution to system errors.

Point 9: What can you say about the thermal transport mechanism within the composite? Is it possible to be through BNNP percolation network?

Response 9: Thanks for your nice point. The heat transfer of composite materials with low filler content has always been a problem. In our research, The BNNPs in this composite material has a regular arrangement in the in-plane direction, which can form a heat conduction path on the microscopic level. It can be seen from the SEM images in Fig.2 a and b that the BNNPs is tightly connected at the surface of the PE film, which, from another aspect, can be seen as percolation network in the composite.

Round 2

Reviewer 3 Report

The authors changed the fitting of Fig S5, but since there is a large deviation of the raw data from linear distribution, an explanation/discussion of the errors affecting these results should be included on figure caption.
